# Novel Phenotypical and Biochemical Findings in Mucolipidosis Type II

**DOI:** 10.3390/ijms26062408

**Published:** 2025-03-07

**Authors:** Eines Monteagudo-Vilavedra, Daniel Rodrigues, Giorgia Vella, Susana B. Bravo, Carmen Pena, Laura Lopez-Valverde, Cristobal Colon, Paula Sanchez-Pintos, Francisco J. Otero Espinar, Maria L. Couce, J. Victor Alvarez

**Affiliations:** 1Metabolopathies Platform, IDIS—Health Research Institute of Santiago de Compostela, 15706 Santiago de Compostela, Spain; eines.monteagudo.vilavedra@sergas.es (E.M.-V.); daniel.caiola.candeias@sergas.es (D.R.); giovella123@gmail.com (G.V.); laura.lopez.valvederde@sergas.es (L.L.-V.); cristobal.colon.mejeras@sergas.es (C.C.); paula.sanchez.pintos@sergas.es (P.S.-P.); 2Metabolic Unit, Clinical University Hospital of Santiago de Compostela, Centro de Investigaciones Biomedicas en Red de Enfermedades Raras (CIBERER), European Reference Network for Rare Hereditary Metabolic Disorders (MetabERN), 15706 Santiago de Compostela, Spain; 3Department of Forensic Sciences, Pathology, Gynecology and Obstetrics, Pediatrics, University of Santiago de Compostela, 15706 Santiago de Compostela, Spain; 4Proteomic Platform, IDIS—Health Research Institute of Santiago de Compostela, 15706 Santiago de Compostela, Spain; carmen.pena.pena@sergas.es; 5Redsamid: Red de Salud Materno Infantil y del Desarrollo, Instituto de Salud Carlos III (ISCIII), 20829 Madrid, Spain; 6Paraquasil Platform, IDIS—Health Research Institute of Santiago de Compostela, 15706 Santiago de Compostela, Spain; francisco.otero@usc.es; 7Instituto of Materials—iMATUS, Department of Pharmacology, Pharmacy and Pharmaceutical Technology, School of Pharmacy, Campus Vida, University of Santiago de Compostela, 15872 Santiago de Compostela, Spain

**Keywords:** mucolipidosis II-III, early findings, biochemistry, biomarkers, proteomic studies, lysosomal diseases

## Abstract

Mucolipidosis type II is a very rare lysosomal disease affecting the UDP-GlcNAc N-acetylglucosamine-1-phosphotransferase enzyme, which catalyzes the synthesis of the targeting signal mannose 6-phosphate in lysosomal acid hydrolases. Its deficiency hinders the arrival of lysosomal enzymes to the lysosome, diminishing the multiple degradations of components that cells need to perform. Due to the low prevalence of this condition, available information is scarce. This article aims to deepen the understanding of the disease; clinical, biochemical, and proteomic data are analyzed. Three patients have been identified presenting *GNPTAB* pathogenic variants using whole exome sequencing. A biochemical profile for these patients has been carried out through quantification of glycosaminoglycans in urine samples and enzymatic analysis in dried blood spot (DBS) samples. Quantitative proteomic studies were performed. Results show how enzymatic assays in DBS can be used to diagnose this disease both during the neonatal period or in patients of more advanced age. Increased levels of acid sphingomyelinase, alpha-iduronidase, iduronidate 2-sulfatase, alpha-N-acetyl glucosaminidase, and beta-glucuronidase are found. Conclusion: this biochemical method could potentially improve early diagnosis. Proteomic data supporting these results reveal disrupted biochemical pathways, including the degradation of dermatan sulfate, heparan sulfate, and cellular cholesterol trafficking.

## 1. Introduction

Mucolipidosis (ML) II (MIM#252500) is an autosomal recessive lysosomal storage disorder caused by mutations in the *GNPTAB* gene producing N-acetylglucosamine-1-phosphotransferase (GlcNAc-PTase, EC 2.7.8.17) deficiency. The final consequence is a defect in the mannose 6-phosphate (M6P) targeting signal function, which generates the hypersecretion of lysosomal enzymes out of cells. Their respective substrates gather inside lysosomes in several tissues (fibroblasts, secretory organs, and connective tissue are severely affected); their accumulation along with the altered traffic of lysosomal enzymes produces the clinical signs and symptoms of these diseases [1]. These alterations are cell type and tissue specific, predominantly affecting fibroblasts and other mesenchymal cells; in other tissues, such as the brain, Man-6-P-independent mechanisms have been found to direct lysosomal enzymes to lysosomes and partially replace its function [2,3] but cannot fully prevent neurologic manifestations [4]. Also, M6P dependency for lysosomal enzyme trafficking could differ between enzymes [1].

ML is a rare disease with an estimated prevalence of 0.22–2.7 per 100,000 live births [5,6]. The combined birth prevalence is estimated to be between 1/37,000 and 455,000 worldwide, whereas the reported birth prevalence for ML II is estimated to be between 1/123,000 and 2,000,000 in Europe (www.orpha.net, accessed 3 January 2025). A founder effect has been reported in the Saguenay-Lac-St-Jean region of Quebec where a birth prevalence is estimated at 1/6000 [7].

ML II is considered a multisystemic disease mainly characterized by skeletal abnormalities (dysostosis multiplex), which cause significant growth restriction with short stature and low weight. Other frequent signs and symptoms (>10%) include dysmorphic facial features, developmental delay, and restricted joint range of motion [8]. Gingival hyperplasia is a distinctive feature of this disease. Cardiac involvement with heart valve disease, cardiomyopathy, or left ventricular hypertrophy can also be found, as well as respiratory problems due to mucosal thickening and stiffening of the thoracic cage. Other occasional clinical features include hepatosplenomegaly, abdominal wall defects (diastasis recti, umbilical and inguinal hernia), recurrent respiratory infection, and hypotonia [1,9]. Even though most lysosomal diseases have a nonspecific neonatal presentation, in most cases, symptoms are already present at birth [8] and the phenotype is already distinctive at this time [1] and can help to establish a diagnostic hypothesis. ML II patients can present early symptoms such as hyperparathyroidism right after birth [1,10,11] and radiological evidence may be found even in the fetal period pointing to ML II [12,13,14]. However, due to the low prevalence of the disease, data on its clinical presentation, natural history, and biochemical profile are scarce.

There is currently no specific treatment for ML II and ML III α/β and the prognosis of the disease is variable but generally very poor, with a median survival of 5 years for the ML II phenotype [8]. Most patients die from pulmonary and cardiac complications. Research on potential therapies is currently underway, with pending results, which may open new horizons for these patients.

Identifying these patients remains a significant challenge. In the absence of a biochemical marker, most patients are lately diagnosed by genetic study, probably later than expected in other lysosomal diseases. However, abnormalities in lysosomal enzyme activity in plasma and dried blood spots (DBS) can be detected using Liquid Chromatography-Mass Spectrometry (LC-MS/MS), without a specific profile. Altered glycosaminoglycan (GAGs) levels can be found as well in serum/plasma, DBS, and urine specimens [1,15,16]. In this study, enzymatic and proteomic tests are carried out to shed light on poorly described and understood aspects of the disease.

## 2. Results

### 2.1. Clinical and Genetic Characteristics

Three patients with a confirmed diagnosis of ML II were enrolled in this study and followed up for an average of 24 months. The demographic, anthropometric, and main clinical characteristics of the patients are summarized in Table 1.

Patient 1 is a 5-year-old female. Family history included a previous miscarriage due to severe fetal skeletal dysplasia without fetal genetic study. In the third trimester of gestation, she was diagnosed with intrauterine growth restriction. She was born full term, at 37 + 5 weeks of gestational age (GA); at birth, she was admitted to the Neonatal Unit due to low weight (Table 1). During her admission, cardiac anomalies consisting of patent ductus arteriosus and left ventricular diastolic dysfunction were observed. Moreover, she presented left clubfoot and bilateral hip dysplasia.

She was later diagnosed at 5 months of life by clinical whole-exome sequencing (WES), a frameshift variant with premature stop codon [c.738del, p.(Lys246Asnfs*21)] and a deletion including the last two exons of the GNPTAB. (NM_024312.5). This information has been communicated to the Clin Var database (SUB15044432).

As for her skeletal dysplasia, imaging tests were performed showing a cranial deformity (Figure 1) with triangular morphology of the anterior portion of the skull and craniosynostosis of the metopic suture, with parietal and temporal bone widening. Anomalies in the thorax with a widening of costal metaphysis were also incidentally found during radiographic procedures (Figure 2). Over the first year of life, she required a left Achilles tenotomy and arthrography of both hips, with adductor tenotomy.

Over the years, she has developed mild concentric hypertrophy of the left ventricle, severe at the septal level, with no evidence of left ventricular outflow tract obstruction. She also presents heart valve disease, with moderate mitral insufficiency and thickened aortic valve leaflets leading to mild insufficiency, as well as mild pulmonary insufficiency. Ophthalmologic and otorhinolaryngologic evaluations showed corneal turbidity and adenoid hypertrophy with significant obstruction of the choanae, as well as hypertrophy of the pharyngeal mucosa.

On physical examination, she has progressively developed some distinctive features such as infiltrative facies and gingival hyperplasia. She maintains significant growth retardation and weight stagnation (Table 1).

During early childhood, she presented recurrent respiratory infections, so at 4 years of age immunological studies were carried out. Results showed a possible defect in humoral immunity, with evidence of IgM deficiency (IgM < 5 mg/dL).

Regarding her neurodevelopment, there is currently a motor impairment and a slight delay in language development. She can sit without support, but never achieved autonomous ambulation or could stand by herself. At a cognitive level, however, she presents normal development, being able to understand simple sentences, performing symbolic play, and building sentences of two or three words, sometimes with low intelligibility. She suffers from minor feeding difficulties and receives treatment with levothyroxine, due to non-autoimmune hypothyroidism, and captopril. The patient is under multidisciplinary follow-up.

Patient 2 is a 2-year-old male. There is no relevant prenatal history, he was born at 37 + 2 weeks of gestational age. Since birth, he presented subtle dysmorphic features, gingival hyperplasia, as well as altered phospho-calcium metabolism. In the first initial hours of life, he was admitted to the Neonatal Unit because he presented jaundice and blood tests revealed thrombocytopenia with non isoimmune hyperbilirubinemia. At 13 days of life, he had to be admitted again because hypophosphatemia was observed; further studies showed elevated PTH (Table 1) and alkaline phosphatase, being diagnosed with hyperparathyroidism.

During the first two months of life, brachycephaly and trigonocephaly were observed, along with delayed motor development and hypotonia. Imaging tests (Figure 1) showed severe brachycephaly and trigonocephaly with craniosynostosis; due to progressive worsening of his bone alterations, surgery—consisting of cranial remodeling—was performed at 7 months of life. Other skeletal anomalies characteristic of the pathology were also found (Figure 2). The results of genetic studies were obtained at that time confirming the ML II diagnosis. Clinical WES showed a frameshift variant in homozygosis (c.3503_3504del) in the *GNPTAB* gene classified as pathogenic, which is the most prevalent variant of this disease worldwide [8,17].

Since the patient showed evolutionary feeding difficulties with episodes of choking and weight stagnation (Table 1), it was decided to start enteral nutrition through a nasogastric tube at one year of life. A gastrostomy was later performed; as in the previous surgery, a difficult airway was observed, requiring the use of a laryngeal mask and intubation through fibro bronchoscopy.

Although his motor development is delayed, his neurodevelopment at social and cognitive levels keeps progressing. His joint contractures limit his ability to extend his knees and arms, but he can manipulate objects with both hands. Currently, he does not receive any pharmacological treatment.

Patient 3 is an 11-month-old male with a severe neonatal onset of the disease. During prenatal ultrasound evaluations, a delayed intrauterine growth was observed; due to fetal distress risk, a cesarean section was performed at 29 + 4 weeks of gestational age after receiving lung maturation with corticosteroids and neuroprotection with magnesium sulfate. He needed immediate admission to the NICU (Neonatal intensive care unit) due to respiratory distress requiring endotracheal intubation.

During admission, the patient was extubated in the first 24 h of life but there was a persistent need for oxygen and non-invasive respiratory support, probably related to bronchopulmonary dysplasia without significant response to medical treatment with inhaled corticosteroids and oral spironolactone. A thoracic CT (Computed Tomography) scan, at two months of age, showed significant bilateral pulmonary involvement with parenchymal fibrosis and volume loss. Along with the chronic lung disease, abdominal wall defects, osteopenia, and dysmorphic features were also identified. The passage of meconium was delayed until the eighth day of life. He presented diastasis recti, with bilateral reducible inguinal hernias and right hydrocele.

Dysmorphic features with abnormal cranial morphology have been striking since birth, with severe osteopenia and spontaneous distal fracture of the right tibia diagnosed at 5 weeks of life. Due to secondary neonatal hyperparathyroidism (Table 1), he was treated with vitamin D, monosodium phosphate, calcium gluconate, and later bisphosphonates. He also presented difficulties in sucking and swallowing from the first days of life, requiring a nasogastric tube for feeding. He was diagnosed at 4 months of age by clinical WES; the patient presented two pathogenic variants in heterozygosis (c.2956C>A and c.3503_3504del), affecting the *GNPTAB* gene.

During the first months of life, he needed several hospital admissions due to respiratory failure, requiring continued support with non-invasive mechanical ventilation. At 8 months of life, it was decided to perform a tracheostomy along with a gastrostomy and surgical correction of the abdominal wall defects. He currently presents a stationary clinical situation at the respiratory level, with occasional decompensation, maintaining mechanical ventilation in spontaneous mode.

In terms of neurodevelopment, significant generalized hypotonia persists and he is not able to consistently hold his head up.

From an immunological point of view, patient 2 has presented mild respiratory infections while patient 3 has been asymptomatic. Immunological studies were performed in both patients confirming the presence of humoral alterations, with a profile consistent with IgM production deficit for patient 2 (IgM 8 mg/dL) and IgG values at the low end of the normal range for patient 3 (IgG 262 mg/dL).

### 2.2. Biochemical Characteristics: GAG and Enzymatic Activity Profile, Plasma Proteomic Analysis

In urine samples for GAG quantification (neonatal samples obtained for routine newborn screening), we observed patient 1 had elevated dermatan sulfate (DS), heparan sulfate (HS) and chondroitin sulfate (CS), in patient 2 all the values were normal, and patient 3 showed elevate DS and HS. However, in the diagnostic samples, all the values obtained were normal in all 3 patients. The results are shown in Table 2.

The enzymatic analysis showed high values of acid sphingomyelinase (ASMD) EC 3.1.4.12, alpha-iduronidase (IDUA) EC 3.2.1.76, iduronidate 2-sulfatase (IDS) EC 3.1.6.13, alpha-N-acetyl glucosaminidase (NAGLU) EC 3.2.1.50, and beta-glucuronidase (GUSB) EC 3.2.1.31 in the neonatal samples and diagnostic samples of the three patients, with the only exception being the arylsulfatase B (ARSB) EC 3.1.6.12 value of patient 1 in the neonatal sample. The results are shown in Table 3.

Alpha-galactosidase (GAA) (EC 3.2.1.22) enzyme analysis showed elevated values only in diagnostic samples, with normal results in neonatal samples. The other enzymes (galactocerebrosidase-GALC (EC 3.2.1.46), alpha-glucosidase-GLA (EC 3.2.1.20), beta-glucosidase-GBA (3.2.1.21), galactose 6-sulfatase-GANLS (EC 2.5.1.5), and beta-galactosidase-GLB1 (EC 3.2.1.23) presented normal values, with a few isolated results slightly above or below the normal level. The results are shown in Table 4.

In the following figure (Figure 3), we put all the enzymes that participate in the degradation routes of the glycosaminoglycans DS and HS. We marked the enzymes not analyzed in the DBS samples of ML whose results have been elevated in all the analyzed enzymes.

### 2.3. Quantitative Proteomic Analysis by SWATH

Samples from ML II patients, both at birth and diagnosis, and samples from healthy neonates were analyzed using SWATH proteomic technique.

In neonatal samples of patients affected by ML II, the upregulated proteins are shown in Table 5. The elevated proteins of healthy newborn samples when compared to affected neonates are shown in Table 6.

In Figure 4, a volcano plot from the SWATH-MS quantitative proteomics analysis and the results from the string tool analysis are shown.

In the proteins upregulated in healthy samples compared with neonatal disease samples, we found PPBG and HEXB proteins, which are related to lysosomal activity, and CEPT protein, which is related to cholesterol transport.

In diagnostic samples compared with healthy samples, we found 36 dysregulated proteins (see Figure 5). The proteins upregulated in ML II patients at the time of diagnosis are shown in Table 7; the downregulated proteins in ML II patients are shown in Table 8.

Among these dysregulated proteins, we found the following interesting proteins: APOD protein related to cholesterol; G3P, LDHA, and LDHB proteins related to carbon energy generation; and HEXB protein related to lysosomal activity.

Further on, when we compared samples from healthy neonates vs. affected patients (neonatal period and at the time of ML II diagnosis), we found 32 dysregulated proteins (see Figure 6). The proteins downregulated in ML II patients are shown in Table 9; the upregulated proteins in ML II patients are shown in Table 10.

In the comparison between healthy neonates and samples from affected patients, we found proteins previously found such as HEXB, G3P, and CATG.

When we compared the dysregulated proteins, we found interesting common proteins (Table 11 and Figure 7), such as HEXB, elevated in healthy individuals and downregulated in patients (both during the neonatal period and at the time of diagnosis).

## 3. Discussion

The overall goal of the present study was to investigate the phenotype, metabolomic, enzymatic, and proteomic profiles involved in MLII patients and to explore the clinical utility. From a clinical point of view, patient 3 presented a severe neonatal form of the disease while patients 1 and 2 presented later and more moderate forms.

The bone involvement of our patients is similar to that which is previously described in the literature [1,9]. All three patients showed skeletal malformations detected during the first year of life, and cranial malformation (Figure 1) seems to be a very consistent finding in these patients [1,18]. Deciding to perform a craniotomy may be questionable [9] since craniosynostosis cannot be considered to be unequivocally caused by bone involvement [14]. Thoracic alterations in the ribs are subtle but may be an incidental finding of interest.

In two of the three patients, gingival hyperplasia and hypoparathyroidism were also detected as an early distinctive feature. Although ML II is a rare disease, early biochemical findings together with skeletal alterations (especially intrauterine or spontaneous fractures and cranial malformation) and characteristic facial features may be clinically suggestive of the disease.

Respiratory tract involvement is described in ML II [19]; however, we consider the chronic lung disease shown by patient 3 to be related to his prematurity since this type of pulmonary findings has not been described in any other patient with the disease. It is noteworthy that, as expected, two of the three patients required advanced airway management during surgery due to a complex airway probably related to mucosal thickening and deposit of metabolic substrate [1]. A potentially life-threatening airway distortion has been previously described [20] but more knowledge of the possible causes and their management is needed.

Two patients required gastrostomy placement due to their inability to feed. Even so, over time all individuals have presented severe growth problems with failure to thrive. The origin of this growth problem remains to be clarified.

From a neurodevelopmental point of view, a severe delay in the motor development of the three patients has been found, related to hypotonia. None of the patients achieved autonomous ambulation. In contrast, cognitive dysfunction was present but it was not as significant compared with motor impairment. In other patients, significant sensory-motor impairment has also been described, in contrast to much less affected social and communication skills [21]. The evolution followed by our patients shows a delay in the acquisition of cognitive items but with continuous progress. The presence or absence of possible neurodegeneration in the disease seems controversial [2,21,22]. Further studies are needed to elucidate the effect that the loss of the Man-6-P targeting signal has on the central nervous system.

Although recurrent respiratory infections have been reported as part of the disease, the humoral immunity deficit affecting ML II patients is poorly understood. The pathogenesis and clinical implications of these findings remain to be clarified, research in affected mice suggests a dysfunction in M6P transport routes could lead to alterations in B cell functions, while DC and T cell functions remain normal due to M6P-independent targeting pathways [23].

Previously identified pathogenic variants of the disease have been found in patients 2 and 3. Patient 1 presents a new heterozygous variant (c.738del, p.(Lys246Asnfs*21)), which has not been previously described, in combination with a deletion affecting exons 20 and 21 of the gene.

When enzymatic studies are carried out, the analysis conditions replicate the ideal conditions of lysosomes due to the acidic pH that the enzymes need to be functional. This has always made diagnosing the disease challenging, as the location of the enzymes within the cells is not well understood (many are found in the cytoplasm, while others are within the lysosome). As a result, genetic studies have been essential for diagnosing this condition [1,24].

The results obtained from the enzymatic activity of this study do have a special impact on the pathophysiology and diagnosis of ML II. In this case, 12 different enzymes have been used, all of them hydrolases whose site of action is the lysosome. The enzymes found to be significantly high during the analysis (IDUA, IDS, NAGLU, GUSB, and ARSB) are part of the HS and DS degradation pathways (Figure 3). The enzyme ASMD is involved in cholesterol degradation. All these enzymes need a mannose 6-phosphate ligand to reach the lysosome. The data obtained are of great diagnostic value as these alterations are maintained regardless of age and, given their presence since the neonatal period, they constitute a key diagnostic tool for early diagnosis of the disease.

The results obtained for the GAA, GALC, GLA, GANLS, and GLB1 enzymes are not conclusive, as is the case with the GAGs. Some of these tests are mentioned in articles as being able to be used as a suspicion of this disease. The same is said about the elevated IDUA and IDS; however, until now they had not all been studied together [15,16].

Data obtained from proteomic studies, in the comparison of ML II neonates vs. healthy neonates, show elevated levels of several proteins. These include the ODPB protein, which participates in the Krebs cycle, the lysosomal protein CATG cathepsin G, which activates extracellular matrix degradation, the MMP8 metalloprotease, which degrades extracellular matrix [25], and AACT alpha-1-antichymotrypsin, which inactivates cathepsin G. In addition, we also found the VDAC2 protein, which binds to various lipids such as the sphingolipid ceramide, among others, and cholesterol sterols [26].

In the healthy neonate vs. ML II neonate comparison, the CETP protein stands out. This protein is involved in lipid transfer, including the transfer of cholesterol and triglyceride esters, allowing the movement of cholesterol esters. In addition, it also regulates reverse cholesterol transport; by this process, excess cholesterol is removed from peripheral tissues [27].

The proteins we found that were elevated, together with the enzyme activities that were also elevated, show us a process of activation of matrix degradation to extract sugars from GAGs for energy production [28,29]. Due to a malfunction of lysosomal trafficking, caused by M6P deficiency, lysosomal hydrolases are not able to reach the lysosome, resulting in a large increase in the production of these enzymes to compensate and to obtain energy from sugars. These processes are common in situations of absence of nutrients in the cell (autophagy situations) and could explain why GAGs may be found normal or minimally elevated.

Similarly, the finding of low CEPT in patients with MLII compared to healthy individuals reveals difficulties in cholesterol transport and esterification; data obtained from ASMD activity also support this disturbance in cholesterol utilization.

It is also important to mention the alteration found in the LAMP2 protein. This lysosomal transmembrane protein helps to internalize hydrolases [30,31]. It has been found to be high in neonatal ML II but its relevance disappears in samples from older patients (at diagnosis). This loss of function has been previously described by Takanobu Otomo et al. [32].

PPGB proteins are protective proteins that appear to be essential for both beta-galactosidase and neuraminidase activity. PPGB exerts a protective function necessary for cell stability and could help prevent the generation of gangliosides, molecules that accumulate in MLII, and other lysosomal diseases [22].

The HEXB protein is always higher in the healthy group than in the group affected by ML II. Authors such as Whelan et al. [33] found the activity of this enzyme to be deficient in the cells but elevated in plasma.

The low number of patients is the main limitation of this research. Due to the very low prevalence of the disease and its high mortality, it is difficult to identify and include patients in research projects. In addition, the proteomic studies have been performed on DBS samples. Although the results are very relevant, we consider they should be validated on liquid samples.

## 4. Materials and Methods

### 4.1. Patients

Three patients, two male and one female, have been identified in our center presenting GNPTAB variants. Written consent for publication of patient data was obtained from the families of all individuals included.

Demographic information and clinical data including fetal, early, and evolutionary clinical characteristics of the patients were collected. Laboratory variables performed include phosphocalcic metabolism study (PTH, calcium, phosphorus, alkaline phosphatase, vitamin D) and immunological profile including quantification of immunoglobulins IgA, IgG (IgG1, IgG2, IgG3, IgG4) and IgM, total number of lymphocytes, CD4 and CD8 lymphocytes, B-CD19 lymphocytes, B-CD20 lymphocytes, memory B lymphocytes, and NK lymphocytes (NK-CD19, NK-CD56, NK-CD57) developed in our local laboratory. The normal range for the biochemical variables according to age: PTH 15–60 pg/mL; Alkaline phosphatase 1 week–6 months < 1076 U/L, 7 months–1 year < 1107. Normal range for immunoglobulins according to age: IgM 3–6 months 5–50 mg/dL, 6–12 months 8–70 mg/dL, 12–24 months 10–100 mg/dL, 2 years–6 years 50–180 mg/dL; IgG 3–6 months 170–560 mg/dL, 6–12 months 200–670 mg/dL, 12–24 months 300–1160 mg/dL, 2 years–6 years 400–1100 mg/dL.

### 4.2. Methods

#### 4.2.1. Molecular Diagnosis

Genomic DNA was obtained from blood samples and variants were identified using WES, performed as part of routine diagnostic protocols. WES was performed using Illumina NextSeq (Illumina, Inc., San Diego, CA, USA). The library preparation method used was the TruSeq DNA Library Prep Kit (Illumina, Inc.), and the capture of regions of interest was performed using IDT xGen Exome ResearchPanel (Integrated DNA Technologies, Inc., Coralville, IA, USA).

#### 4.2.2. Biochemical and Enzymatic Profile

To carry out the studies of GAG and enzymatic activity, DBS and dry urine spot (DUS) samples were selected from the 3 clinical cases described above. Neonatal samples for routine newborn screening were selected (collected 24 h after the first lactation); samples collected as part of this study of a possible lysosomal disease (diagnosis) of each patient were also used.

GAG levels were quantified in DUS and enzymatic analysis was performed in DBS specimens using LC-MS/MS. The following enzymes were studied in the DBS samples: ASMD, IDUA, IDS, NAGLU, ARSB, GUSB, GALC, alpha-glucosidase, alpha-galactosidase, beta-glucosidase, galactose 6-sulfatase, and beta-galactosidase. Subsequent proteomic analyses were also conducted at our laboratory.

#### 4.2.3. GAG Analysis

For the quantification of urinary GAGs, we adopted a method previously described [34]. Briefly, DUS from urine samples and calibrators (DS, HS, CS, and Creatinine) was subjected to methanolysis for 1 h at 65 °C in a 96-well plate. The resulting supernatant was evaporated under a nitrogen stream and subsequently resuspended in a solution containing the internal standard. LC-MS/MS was performed using an AB Sciex API 4500 QTRAP in positive ion mode to quantify the three types of GAGs and Creatinine. Table 2 presents the results of the GAG analysis in the neonate period and at diagnosis from the three patients.

#### 4.2.4. Enzymatic Analysis

Further biochemical characterization was performed through enzymatic analysis. For this purpose, two kits from Revvity (Waltham, MA, USA) were used. The first, the NeoLSD™ MSMS Kit (cat. 3093-0010), allows for the multiplex quantitative measurement of the activity of several enzymes: acid-β-glucocerebrosidase (Gaucher disease), acid-sphingomyelinase (acid sphingomyelinase deficiency), acid α-glucosidase (Pompe disease), β-galactocerebrosidase (Krabbe disease), α-galactosidase A (Fabry disease), and α-L-iduronidase (Mucopolysaccharidosis type I disease—MPS I). The method was adapted to convert flow injection into gradient measurement [35]. Quantification was performed using an AB Sciex API 4500 QTRAP in positive ion mode using a UPLC column from Waters (X Select CSH C18 3.5 µm, 2.1 × 50 mm (Cat. 186005255) held at 55 °C. The flow rate was 0.7 mL/min. Solvent A was 30% acetonitrile with 0.1% formic acid, and solvent B was 65% isopropanol/35% acetonitrile with 0.1% formic acid. The gradient is shown in Table 12.

The second kit (cat. 4416-0010) allows for the multiplex quantitative measurement of the activity of: iduronate 2-sulfatase (MPS II), α-N-acetylglucosaminidase (MPS IIIB), galactose 6-sulfatase (MPS IV A), β-galactosidase (MPS IV B), N-acetylgalactosamine (MPS VI), β-glucoronidase (MPS VII), and tripeptidyl peptidase (CLN2). Adaptations were also made for gradient measurement [35]. The same equipment, UPLC column, and gradient solvents were used as in the NeoLSD kit. The gradient used for MPS measurement is shown in Table 13.

#### 4.2.5. Proteomic Studies

Proteomic studies are quantitative studies using SWATH (sequential window acquisition of all theoretical fragment ion spectra), an information-dependent acquisition (IDA) technique was used. A human library containing 2150 proteins has been used, from which 1.021 have been quantified.

### 4.3. Experimental Design and Statistical Analysis

Proteomic Analysis by TripleTOF 6600 using liquid chromatography–tandem mass spectrometry (LC-MS/MS): (see Appendix A for further information).Protein extraction: Protein from Dried Blood Samples was extracted by incubating the paper in 100 μL of 100 mM ammonium bicarbonate at room temperature for 1 h. The sample was centrifuged for 20 min at 13,000× *g*, and the supernatant was transferred to a new tube. Then the protein was precipitated by the MeOH/CHCl_3_ method, and the protein concentration was measured using an RC DC™ Protein Assay (reducing agent and detergent compatible) (BioRad, Hercules, CA, USA).Protein Digestion: For protein identification, equal amounts of protein from each sample (*n* = 3 per group and 4 healthy controls) were loaded onto a 10% SDS-PAGE gel. The resulting condensed protein bands [36,37] underwent gel digestion using Trypsin and were processed, as previously described in the Appendix A by our group [28].Protein Quantification by SWATH-MS Analysis [28,38,39,40]: To build the MSMS spectral libraries, peptide solutions were analyzed by shotgun data-dependent acquisition (DDA) using micro-liquid chromatography–tandem mass spectrometry (LC-MS/MS), as described in the Appendix A and previously by our group. The MSMS spectra of the identified peptides were then used to generate the spectral library for SWATH peak extraction using the add-in for PeakView Software (version 2.2, Sciex), MSMS^ALL^ with SWATH Acquisition MicroApp (version 2.0, Sciex). Peptides with a confidence score >99% (obtained from the ProteinPilot database search) were included in the spectral library. For relative quantification by SWATHMS analysis, SWATH-MS acquisition was performed on a Triple TOF 6600 LC–MS–MS system (Sciex) using SWATH mode. The acquisition mode consisted of a 250 ms survey MS scan from 400 to 1250 *m*/*z*, followed by an MSMS scan from 100 to 1500 *m*/*z* (25 ms acquisition time) of the top 65 precursor ions from the survey scan, for a total cycle time of 2.8 s. The fragmented precursors were then added to a dynamic exclusion list for 15 s. Any singly charged ions were excluded from the MSMS analysis. Targeted data extraction from the SWATH MS runs was performed by PeakView v.2.2 (Sciex, Redwood City, CA, USA) using the SWATH-MS Acquisition MicroApp v.2.0 (Sciex, USA). Data were processed using the spectral library created from DDA. SWATH-MS quantization was attempted for all proteins in the ion library that were identified by ProteinPilot^TM^ 5.0.1 with a false discovery rate (FDR) < 1%. PeakView computed an FDR and a score for each assigned peptide based on the chromatographic and spectra components: only peptides with an FDR < 1%, 10 peptides, and 7 transitions per peptide were used for protein quantization. The integrated peak areas were processed by MarkerView software version 1.3.1 (Sciex, USA) for a data-independent method for relative quantitative analysis. A most likely ratio normalization was performed to control for possible uneven sample loss across the different samples during the sample preparation process [41,42]. Unsupervised multivariate statistical analysis using PCA was performed to compare data across samples.

### 4.4. Functional and Pathway Analysis

Pathway analysis was performed using Reactome (https://reactome.org/, accessed on 20 July 2023), which applies a statistical (hypergeometric distribution) test to determine whether specific pathways are over-represented (enriched) and produces a probability score, which is corrected for FDR using the Benjamini–Hochberg method. The most enriched pathways were represented using Reactome pathway diagrams. Protein interactions were evaluated using STRING (https://string-db.org/, accessed on 25 July 2023), applying a minimum required interaction score of PPI = 0.9 (protein–protein interaction) and an FDR < 0.05. Venn diagrams were generated using http://www.interactivenn.net/, accessed on 2 June 2023) and box plots using GraphPad Prism 9. Statistical analyses were performed using Markerview v 1.3.1 or Scaffold software v 5.2.2. Volcano plots and box plots were generated using GraphPad Prism 9.0.0.

## 5. Conclusions

The diagnostic process of ML is complex due to its multiorgan involvement; the diagnostic methods up to now have not been properly established because they are indirect enzymatic methods. Since procedures such as the measurement of M6P-containing hydrolases depend on which one is selected and studied, results can be inconclusive. In this article, we describe a group of enzymes that should always be elevated in this type of disease, such as ASMD, and HS- and DS-degrading enzymes.

Multiplex enzyme studies, which are increasingly used in LC-MS/MS techniques, can facilitate diagnosis by simply measuring ASMD and IDUA. In addition, proteomic data are sufficiently robust to demonstrate these enzymes are elevated due to compensatory mechanisms for energy production in cells. This alternative process to generate energy produces negative side effects on cells that become chronic over time. Certain proteins found in this research may be of relevance for the design of future studies and to shed light on the pathology of the disease and the molecular processes involved.

## Figures and Tables

**Figure 1 ijms-26-02408-f001:**
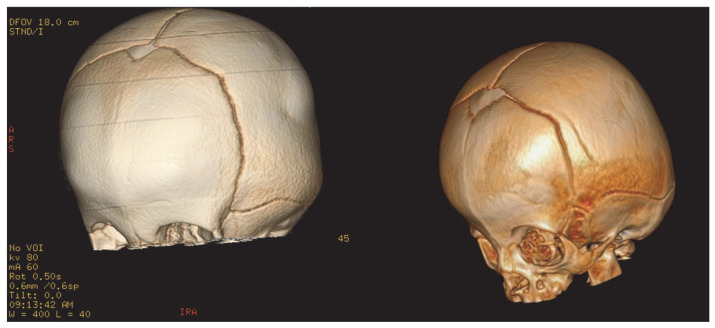
Alterations in cranial morphology. Cranial deformity with trigonocephaly and early closure of the metopic suture, patients 1 and 2.

**Figure 2 ijms-26-02408-f002:**
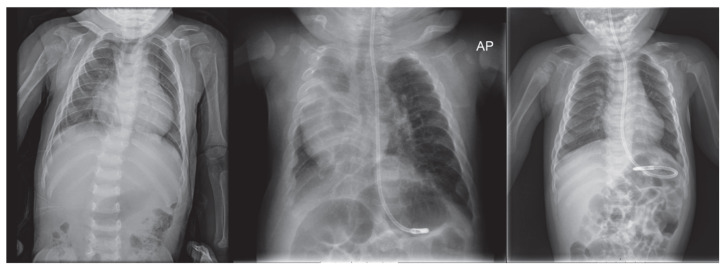
Skeletal findings of the three patients. Deformity of costal arches with widening of the costal metaphysis, patients 1, 2, and 3. AP: anteroposterior protection from X-rays.

**Figure 3 ijms-26-02408-f003:**
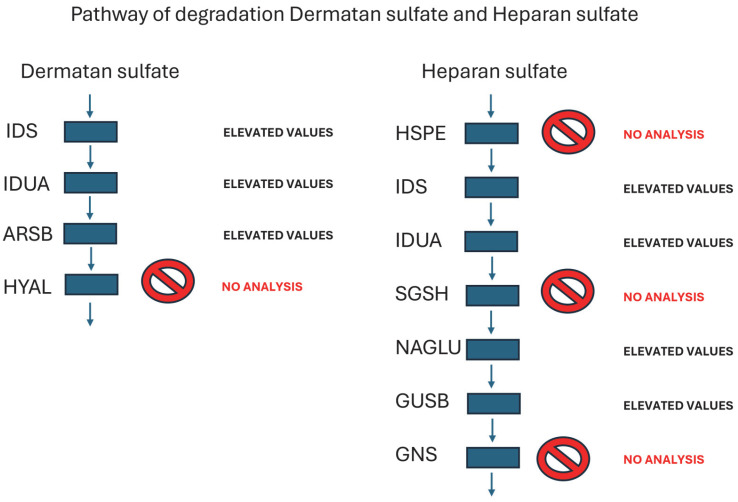
Biochemical routes for degradation of Heparan sulfate and Dermatan sulfate. Enzymes analyzed: IDS—iduronidate 2-sulfatase, IDUA—alpha-iduronidase; ARSB—arylsulfatase B, NAGLU alpha-N-acetyl glucosaminidase; GUSB—beta-glucuronidase. Enzymes not analyzed: Heparanase—HSPE, N-sulfoglucosamine sulfohydrolase—SGSH, N-acetylglucosamine-6-sulfatase—GNS, Hyaluronidase—HYAL.

**Figure 4 ijms-26-02408-f004:**
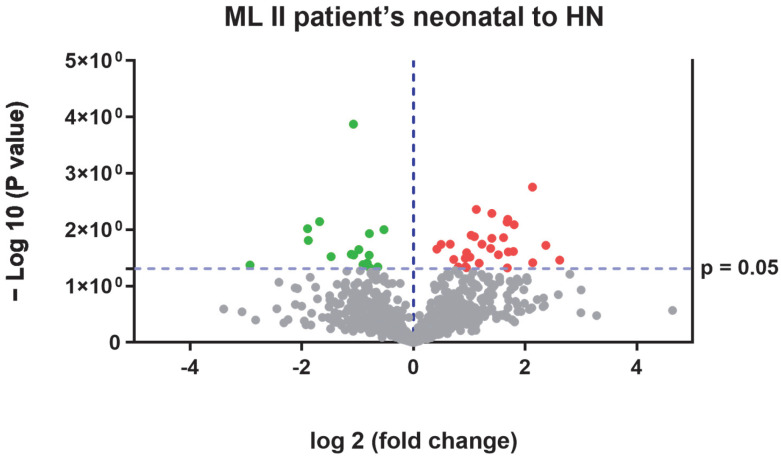
Volcano plot obtained from SWATH-MS quantitative proteomics analysis. The red dots represent proteins upregulated in ML II patients vs. neonatal healthy samples and green dots represent proteins upregulated in neonatal healthy samples vs. ML II patient samples.

**Figure 5 ijms-26-02408-f005:**
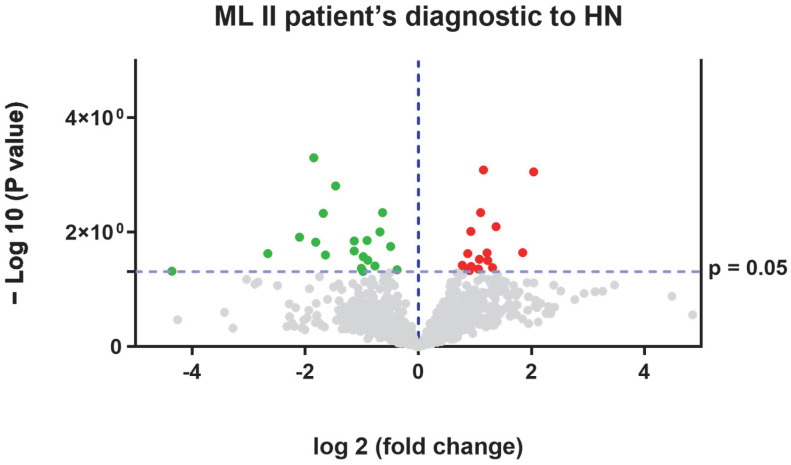
Volcano plot obtained from SWATH-MS quantitative proteomics analysis. The red dots represent proteins upregulated in ML II patients at diagnosis vs. neonatal healthy samples and green dots represent proteins upregulated in neonatal healthy vs. ML II patients.

**Figure 6 ijms-26-02408-f006:**
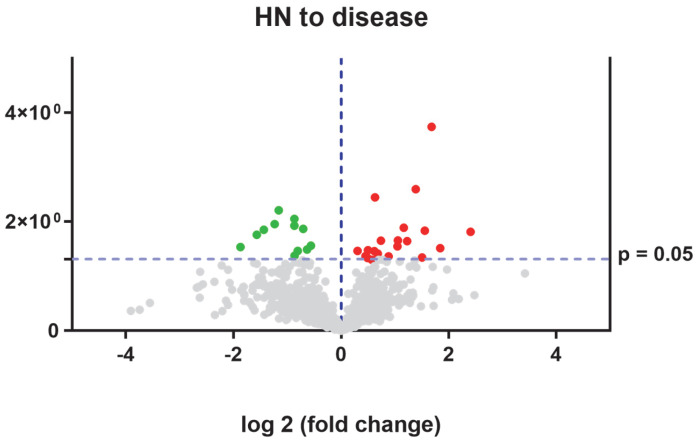
Volcano plot obtained from SWATH-MS quantitative proteomics analysis. The red dots represent proteins upregulated in healthy vs. disease samples and green dots represent proteins upregulated in disease ML II patients.

**Figure 7 ijms-26-02408-f007:**
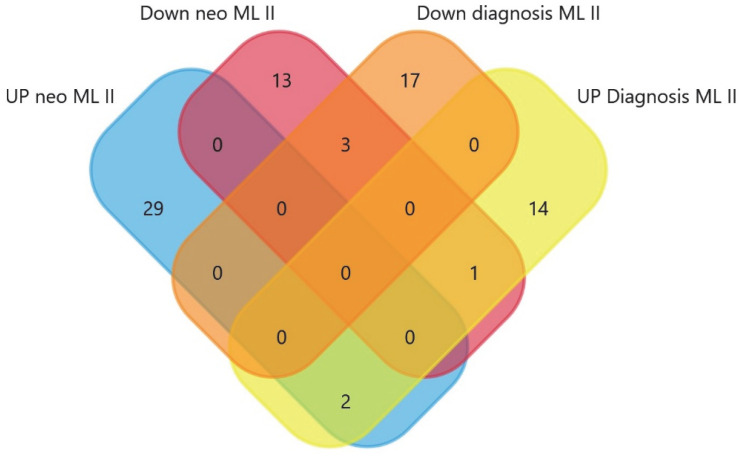
Venn diagram between the dysregulated proteins in different comparisons.

**Table 1 ijms-26-02408-t001:** Summary of demographic, anthropometric, and clinical characteristics of patients 1, 2, and 3.

Pat.	Patient 1	Patient 2	Patient 3
Gender	Female	Male	Male
Ethnicity	Caucasian	Caucasian	Caucasian
GA (weeks)	37 + 5	37 + 2	29 + 4
Weight (g) and length (cm) at birth	2.020 (p2) 47 (p23)	2.480 (p13) 47.2 (p20)	900 (p2) 36 (p4)
Current age (y)	5	2	0.9
Current weight (g) and length (cm)	7.460 (*p* < 1) 68 (*p* < 1)	7.500 (*p* < 1) 67.5 (*p* < 1)	6.670 (*p* < 1) NA
PTH value (pg/mL)	NA	194	138
Alkaline phosphatase (UI/L)	553	2.687	870
Cardiac findings	Present	Present	Present
Abdominal wall defects	Absent	Present	Present
Recurrent respiratory infections	Present	Absent	Absent
Long bone abnormalities	Present	NA	Present
Hypotonia	Present (sits without support, unable to stand)	Present (sits with support, cephalic control difficulties)	Present (unable to hold his head up)
Language development	Simple verbal language	Limited lexical repertoire, use of non-verbal language (gestures)	Non-verbal language (gestures)
Surgical procedures	Tenotomy	Gastrostomy, cranial remodeling	Gastrostomy, tracheostomy, surgical correction of the abdominal wall defects
Pharmacological treatment	Levothyroxine, captopril	None	Omeprazole

GA: gestational age; NA: not available; y: years; PTH: parathyroid hormone.

**Table 2 ijms-26-02408-t002:** Urine GAGs measured by LC-MS/MS in neonate and patient samples.

Pat.	Age	Creatinine (mmol/dL)	Glycosaminoglycans (mg/mmol Creatinine)		
	Sample Date		DS	HS	CS
Pat. 1	NN	0.13 (0.02–1.06)	**23.1** (8.2–21.4)	**10.0** (1.6–7.2)	**65.6** (18.5–48.9)
	2 y	0.15 (0.13–1.24)	5.1 (0.45–10.6)	3.7 (0.18–1.62)	4.0 (3.6–19.2)
Pat. 2	NN	0.07 (0.02–1.06)	16.3 (8.2–21.4)	6.4 (1.6–7.2)	40.9 (18.5–48.9)
	7.5 m	0.24 (0.01–0.64)	11.2 (2.7–17.1)	4.3 (0.7–4.5)	12.2 (8.6–45.6)
Pat. 3	NN	0.15 (0.02–1.06)	**24.5** (8.2–21.4)	**13.9** (1.6–7.2)	38.2 (18.5–48.9)
	5 m	0.33 (0.01–0.64)	13.2 (2.7–17.1)	9.8 (0.7–4.5)	24.7 (8.6–45.6)

Pat—patient; NN—neonate; y—years; m—months; in parenthesis—normal range. DS—Dermatan Sulfate, HS—Heparan Sulfate, CS—Condroitin Sulfate. High values are marked in bold.

**Table 3 ijms-26-02408-t003:** Enzymatic activities measured (µmol/L/h) by LC-MS/MS in neonate and clinical samples.

Pat.	Age	ASMD	IDUA	IDS	NAGLU	GUSB	ARSB
Pat. 1	NN	**127.6** (3.6–5.0)	**16.7** (1.4–5.2)	**85.9** (10.2–15.4)	**57.9** (8.9–27.2)	**96.7** (51.2–79.2)	20.4 (8.0–27.9)
	5 y	**111.4** (1.4–12)	**34.2** (1.4–12.6)	**172.0** (4.0–30.0)	**98.7** (7.6–38.8)	**175.9** (31.0–90.0)	**54.4** (7.0–27.0)
Pat. 2	NN	**91.5** (3.6–5.0)	**15.1** (1.4–5.2)	**98.8** (10.2–15.4)	**100.0** (8.9–27.2)	**122.2** (51.2–79.2)	**38.4** (8.0–27.9)
	2 y	**150.0** (1.4–12)	**27.1** (1.4–12.6)	**178.0** (4.0–30.0)	**124.0** (7.6–38.8)	**241.0** (31.0–90.0)	**69.6** (7.0–27.0)
Pat. 3	NN	**163.0** (3.6–5.0)	**11.3** (1.4–5.2)	**102.9** (10.2–5.4)	**72.8** (8.9–27.2)	**169.0** (51.2–79.2)	**30.2** (8.0–27.9)
	5 m	**187.0** (1.4–12)	**19.7** (1.4–12.6)	**172.0** (4.0–30.0)	**242.0** (7.6–38.8)	**282.0** (31.0–90.0)	**80.4** (7.0–27.0)

Pat—patient; NN—neonate; y—years; m—months; in parenthesis—normal range; ASMD—acid sphingomyelinase; IDUA—alpha-iduronidase; IDS—iduronidate 2-sulfatase; NAGLU alpha-N-acetyl glucosaminidase; GUSB—beta-glucuronidase; ARSB—arylsulfatase B. High values are marked in bold.

**Table 4 ijms-26-02408-t004:** Enzymatic activities measured (µmol/L/h) by LC-MS/MS in neonate and clinical samples.

Pat.	Age	GALC	GAA	GLA	GBA	GALNS	GLB1
Pat. 1	NN	0.63 (1.0–8.83)	9.6 (2.5–13.1)	1.0 (3.6–15.8)	1.8 (2.9–19.2)	4.4 (3.0–8.6)	15.5 (27.8–60.1)
	5 y	1.5 (0.6–5.6)	**11.5** (1.2–8.6)	5.8 (1.6–10.0)	6.9 (1.6–11.0)	**20.8** (2.3–19.2)	37.4 (9.7–64.0)
Pat. 2	NN	0.63 (1.0–8.83)	8.3 (2.5–13.1)	3.5 (3.6–15.8)	7.7 (2.9–19.2)	3.2 (3.0–8.6)	27.6 (27.8–60.1)
	2 y	1.5 (0.6–5.6)	**9.1** (1.2–8.6)	3.7 (1.6–10.0)	6.7 (1.6–11.0)	15.1 (2.3–19.2)	28.6 (9.7–64.0)
Pat. 3	NN	5.8 (1.0–8.83)	8.1 (2.5–13.1)	4.5 (3.6–15.8)	**23.7** (2.9–19.2)	3.8 (3.0–8.6)	28.1 (27.8–60.1)
	5 m	3.2 (0.6–5.6)	**14.6** (1.2–8.6)	5.4 (1.6–10.0)	7.5 (1.6–11.0)	16.5 (2.3–19.2)	36.5 (9.7–64.0)

Pat—patient; NN—neonate; y—years; m—months; in parenthesis—normal range; GALC—galacto-cerebrosidase; GAA—alpha-galactosidase; GLA—alpha-glucosidase; GBA—beta-glucosidase; GALNS—galactose 6-sulfatase; GLB1—beta-galactosidase. High values are marked in bold.

**Table 5 ijms-26-02408-t005:** Comparison by SWATH analysis between neonatal samples of ML II patients vs. neonatal healthy samples (elevated proteins in ML II neonate patients).

Upregulated Proteins in ML II Neonatal Patients vs. Healthy Neonatal Samples			
Uniprot Code	Protein Code	Protein	Fold Change
P11177	ODPB	Pyruvate dehydrogenase E1 component subunit beta, mitochondrial	6.15
P01011	AACT	Alpha-1-antichymotrypsin	5.17
A0A075B6S2	KVD29	Immunoglobulin kappa variable 2D-29	4.39
P30711	GSTT1	Glutathione S-transferase theta-1	4.38
P13473	LAMP2	Lysosome-associated membrane glycoprotein 2	3.49
P47756	CAPZB	F-actin-capping protein subunit beta	3.46
Q99715	COCA1	Collagen lpha-1(XII) chain	3.24
P63241	IF5A1	Eukaryotic translation initiation factor 5A-1	3.22
P0DJI9	SAA2	Serum amyloid A-2 protein	3.21
Q9Y617	SERC	Phosphoserine aminotransferase	3.19
P01019	ANGT	Angiotensinogen	3.06
P36980	FHR2	Complement factor H-related protein 2	2.87
P22894	MMP8	Neutrophil collagenase	2.64
Q99613	EIF3C	Eukaryotic translation initiation factor 3 subunit C	2.64
P46776	RL27A	60S ribosomal protein L27a	2.61
A0A075B6I1	LV460	Immunoglobulin lambda variable 4–60	2.34
P15880	RS2	40S ribosomal protein S2	2.26
Q9NZI8	IF2B1	Insulin-like growth factor 2 mRNA-binding protein 1	2.18
Q01955	CO4A3	Collagen lpha-3(IV) chain	2.13
P08779	K1C16	Keratin, type I cytoskeletal 16	2.05
P04004	VTNC	Vitronectin	2.02
P08758	ANXA5	Annexin A5	1.94
P13645	K1C10	Keratin, type I cytoskeletal 10	1.93
P11387	TOP1	DNA topoisomerase 1	1.91
Q16775	GLO2	Hydroxyacylglutathione hydrolase, mitochondrial	1.90
P45880	VDAC2	Voltage-dependent anion-selective channel protein 2	1.74
Q8WUM4	PDC6I	Programmed cell death 6-interacting protein	1.68
O43451	MGA	Maltase-glucoamylase, intestinal	1.65
P08311	CATG	Cathepsin G	1.58
Q99808	S29A1	Equilibrative nucleoside transporter 1	1.41
Q9UBW5	BIN2	Bridging integrator 2	1.34

**Table 6 ijms-26-02408-t006:** Comparison of SWATH analysis showing downregulated proteins in healthy neonate samples vs. samples from newborn patients with ML II.

Downregulated ML II Neonatal Patients vs. Healthy Neonatal Samples			
Uniprot Code	Protein Code	Protein	Fold Change
Q32P28	P3H1	Prolyl 3-hydroxylase 1	0.69
P02730	B3AT	Band 3 anion transport protein	0.64
Q14520	HABP2	Hyaluronan-binding protein 2	0.58
Q15181	IPYR	Inorganic pyrophosphatase	0.58
P16403	H12	Histone H1.2	0.58
P04843	RPN1	Dolichyl-diphosphooligosaccharide-protein glycosyltransferase subunit 1	0.56
Q86YW5	TRML1	Trem-like transcript 1 protein	0.53
Q9H299	SH3L3	SH3 domain-binding glutamic acid-rich-like protein 3	0.52
P31947	1433S	14-3-3 protein sigma	0.51
P20851	C4BPB	C4b-binding protein beta chain	0.48
P11597	CETP	Cholesteryl ester transfer protein	0.47
Q96AC1	FERM2	Fermitin family homolog 2	0.46
P0DOY3	IGLC3	Immunoglobulin lambda constant 3	0.36
P07686	HEXB	Beta-hexosaminidase subunit beta	0.31
P00915	CAH1	Carbonic anhydrase 1	0.27
P26639	SYTC	Threonine-tRNA ligase, cytoplasmic	0.27
P10619	PPGB	Lysosomal protective protein	0.13

**Table 7 ijms-26-02408-t007:** Upregulated proteins found in the comparison of SWATH analysis between samples obtained for the diagnosis of ML II patients vs. healthy neonatal sample.

Upregulated Proteins in ML II Diagnosis Patients vs. Healthy Neonatal Samples			
Uniprot Code	Protein Code	Protein	Fold Change
P30405	PPIF	Peptidyl-prolyl cis-trans isomerase F, mitochondrial	4.11
P14314	GLU2B	Glucosidase 2 subunit beta	3.59
O00194	RB27B	Ras-related protein Rab-27B	2.59
P08572	CO4A2	Collagen alpha-2(IV) chain	2.48
Q01955	CO4A3	Collagen alpha-3(IV) chain	2.34
Q04828	AK1C1	Aldo-keto reductase family 1 member C1	2.32
P22105	TENX	Tenascin-X	2.22
O00339	MATN2	Matrilin-2	2.15
P04278	SHBG	Sex hormone-binding globulin	2.11
P55084	ECHB	Trifunctional enzyme subunit beta, mitochondrial	2.09
P05090	APOD	Apolipoprotein D	1.91
P00918	CAH2	Carbonic anhydrase 2	1.90
P10644	KAP0	cAMP-dependent protein kinase type I-alpha regulatory subunit	1.87
Q32P28	P3H1	Prolyl 3-hydroxylase 1	1.83
P06396	GELS	Gelsolin	1.76
Q9Y696	CLIC4	Chloride intracellular channel protein 4	1.71
O43451	MGA	Maltase-glucoamylase, intestinal	1.61

**Table 8 ijms-26-02408-t008:** Downregulated proteins found in the comparison of SWATH analysis between samples obtained for the diagnosis of ML II vs. healthy neonatal samples. FC < 1 protein.

Downregulated ML II Neonatal Patients vs. Healthy Neonatal Samples			
Uniprot Code	Protein Code	Protein	Fold Change
P04406	G3P	Glyceraldehyde-3-phosphate dehydrogenase	0.77
Q15181	IPYR	Inorganic pyrophosphatase	0.71
P37840	SYUA	Alpha-synuclein	0.64
P07195	LDHB	L-lactate dehydrogenase B chain	0.62
P39060	COIA1	Collagen alpha-1(XVIII) chain	0.59
P78417	GSTO1	Glutathione S-transferase omega-1	0.54
P02549	SPTA1	Spectrin alpha chain, erythrocytic 1	0.53
P54727	RD23B	UV excision repair protein RAD23 homolog B	0.51
P13489	RINI	Ribonuclease inhibitor	0.51
P26639	SYTC	Threonine-tRNA ligase, cytoplasmic	0.50
P04040	CATA	Catalase	0.46
P13798	ACPH	Acylamino-acid-releasing enzyme	0.46
P53396	ACLY	ATP-citrate synthase	0.36
P00338	LDHA	L-lactate dehydrogenase A chain	0.32
P07686	HEXB	Beta-hexosaminidase subunit beta	0.31
O43657	TSN6	Tetraspanin-6	0.28
P12259	FA5	Coagulation factor V	0.28
Q14974	IMB1	Importin subunit beta-1	0.23
P28065	PSB9	Proteasome subunit beta type-9	0.16
P02008	HBAZ	Hemoglobin subunit zeta	0.05

**Table 9 ijms-26-02408-t009:** Upregulated proteins found in the comparison of SWATH analysis between samples obtained from healthy neonates vs. samples of ML II patients (from neonatal samples and diagnostic samples).

Upregulated Proteins in Healthy Neonates vs. Samples of ML II Patients (Neonatal Period and Diagnosis)			
Uniprot Code	Protein Code	Protein	Fold Change
P01033	TIMP1	Metalloproteinase inhibitor 1	5.30
P27695	APEX1	DNA-(apurinic or apyrimidinic site) lyase	3.58
P07686	HEXB	Beta-hexosaminidase subunit beta	3.21
P28065	PSB9	Proteasome subunit beta type-9	2.94
Q9UGM5	FETUB	Fetuin-B	2.83
P26639	SYTC	Threonine-tRNA ligase, cytoplasmic	2.61
P39656	OST48	Dolichyl-diphosphooligosaccharide-protein glycosyltransferase 48 kDa subunit	2.34
P0DOY3	IGLC3	Immunoglobulin lambda constant 3	2.24
Q9H299	SH3L3	SH3 domain-binding glutamic acid-rich-like protein 3	2.07
Q9P2J5	SYLC	Leucine-tRNA ligase, cytoplasmic	2.06
P28066	PSA5	Proteasome subunit alpha type-5	1.84
P54920	SNAA	Alpha-soluble NSF attachment protein	1.71
P02549	SPTA1	Spectrin alpha chain, erythrocytic 1	1.67
Q96AC1	FERM2	Fermitin family homolog 2	1.60
Q93009	UBP7	Ubiquitin carboxyl-terminal hydrolase 7	1.60
Q15181	IPYR	Inorganic pyrophosphatase	1.55
P02652	APOA2	Apolipoprotein A-II	1.53
P04843	RPN1	Dolichyl-diphosphooligosaccharide-protein glycosyltransferase subunit 1	1.48
P02730	B3AT	Band 3 anion transport protein	1.41
P17655	CAN2	Calpain-2 catalytic subunit	1.40
P37840	SYUA	Alpha-synuclein	1.37
P04406	G3P	Glyceraldehyde-3-phosphate dehydrogenase	1.24

**Table 10 ijms-26-02408-t010:** Downregulated proteins found in the comparison of SWATH analysis between samples obtained of ML II patients (at birth and diagnosis) vs. healthy neonatal samples.

Downregulated Protein Found in ML II Patients (Neonatal Period and at Diagnosis) vs. Healthy Neonates			
Uniprot Code	Protein Code	Protein	Fold Change
P13796	PLSL	Plastin-2	0.68
P08311	CATG	Cathepsin G	0.64
O43451	MGA	Maltase-glucoamylase, intestinal	0.61
P11387	TOP1	DNA topoisomerase 1	0.57
P04278	SHBG	Sex hormone-binding globulin	0.55
P22105	TENX	Tenascin-X	0.55
P08758	ANXA5	Annexin A5	0.55
Q01955	CO4A3	Collagen alpha-3(IV) chain	0.45
Q99613	EIF3C	Eukaryotic translation initiation factor 3 subunit C	0.42
P01019	ANGT	Angiotensinogen	0.37
P14314	GLU2B	Glucosidase 2 subunit beta	0.34
Q9Y617	SERC	Phosphoserine aminotransferase	0.27

**Table 11 ijms-26-02408-t011:** Common proteins found both in healthy neonates and ML II patients.

**Upregulated Proteins in Neonatal and Diagnosis ML Samples vs. Healthy Neonates**				
**Code Protein**	**Protein Code**	**Protein**	**FC Neo**	**FC Dg**
Q01955	CO4A3	Collagen alpha-3(IV) chain	2.13	2.34
O43451	MGA	Maltase-glucoamylase, intestinal	1.65	1.61
**Downregulated proteins in Healthy Neonates vs. Neonatal and Diagnosis ML Samples**				
**Code Protein**	**Protein Code**	**Protein**	**FC Neo**	**FC Dg**
Q15181	IPYR	Inorganic pyrophosphatase	0.58	0.71
P26639	SYTC	Threonine-tRNA ligase, cytoplasmic	0.27	0.50
P07686	HEXB	Beta-hexosaminidase subunit beta	0.31	0.31
**Upregulated Proteins in Neonatal ML Samples vs. Diagnostic ML Samples Remain Elevated**				
**Code Protein**	**Protein Code**	**Protein**	**FC Neo**	**FC Dg**
P30711	GSTT1	Glutathione S-transferase theta-1	4.38	
P63241	IF5A1	Eukaryotic translation initiation factor 5A-1	3.22	
Q99715	COCA1	Collagen lpha-1(XII) chain	3.24	
Q16775	GLO2	Hydroxyacylglutathione hydrolase, mitochondrial	1.90	
Q9NZI8	IF2B1	Insulin-like growth factor 2 mRNA-binding protein 1	2.18	
P08779	K1C16	Keratin, type I cytoskeletal 16	2.05	

**Table 12 ijms-26-02408-t012:** Gradient used for measurement of enzymatic activities by LC-MS/MS with NeoLSD kit.

Time (min)	0.00	2.50	3.20	3.21	4.00
% B	20	100	100	20	20
Flow (mL/min)	0.70	0.70	0.70	0.70	0.70

**Table 13 ijms-26-02408-t013:** Gradient used for measurement of enzymatic activities by LC-MS/MS with MPS kit.

Time (min)	0.00	0.75	1.00	1.50	1.80	2.15	2.16	4.00
% B	0.50	25	60	75	100	100	0.50	0.50
Flow (mL/min)	0.65	0.65	0.65	0.65	0.65	0.65	0.65	0.65

## Data Availability

The mass spectrometry proteomics data have been deposited to the ProteomeXchange Consortium via the PRIDE partner repository with the data set identifier PXD060270.

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
