# Peer review of "Novel Phenotypical and Biochemical Findings in Mucolipidosis Type II"

_ijms, 2025, doi:10.3390/ijms26062408_

Round 1
Reviewer 1 Report
Comments and Suggestions for Authors
In this paper, the authors characterize the clinical, biochemical and proteomic profiles of three patients with a confirmed diagnosis of mucolipidosis type II, followed up for 24 months. This work is interesting and informative. The text of the article is written in a way that enables the authors' argument to be followed comprehensibly. However, the introduction, discussion and conclusion section of the paper should be a bit more refined to make it a smoother read, e.g. in my subjective opinion the last paragraph of introduction should be one position earlier etc. In addition, there are paragraphs in the discussion which are more suited to the introduction (page 18, lines 427-430), or paragraphs which duplicate information already present (e.g. page 18, lines 242-226) and do not constitute a proper discussion of the results obtained. Figure 7 should also be placed earlier in the text and so on.
Some major and minor issues are listed below:
- Page 2, line 46 – it is: ‘MIM#252500 and MIM#252600 - shouldn't ‘MIM’ be ‘OMIM’ instead?
- Page 2, line 48 - the Enzyme Commission (EC) number of N-acetylglucosamine-1-phosphotransferase should be added. The same recommendation is valid for the other enzymes whose names appear for the first time in the whole text (e.g. lines 109-113)
- Page 2, line 48 – please elaborate on the abbreviation M6P
- Page 2, line 59 - please add the date of access
- Page 3, line 94 and 95 - two different ranges of norms are assigned to the same parameter - IgM - please correct
- Page 3, line 100 - the elaboration of the ‘WES’ abbreviation can be found in the following text (line 202) and the abbreviation itself appears here for the first time - please correct
- Page 3, line 109 – it is ‘using MS/MS – it should rather be: ‘using liquid chromatography-tandem mass spectrometry (LC-MS/MS)’
- Page 3, line 118 - acronyms such as HS, DS should appear early, in line 116
- Page 3, line 123 - add city and state information for Revvity to be consistent with earlier text (e.g. line 101)
- Page 4, line 155 – it is: ‘MeOH/ChCl3’ - it should be: ‘MeOH/ CHCl3’
- Page 4, line 156 – RC-DC kit - full kit name not added - please complete
- Page 4, line 164 - please add a specific literature reference
- Page 6, line 204 – add the name of the database from which the ID for the mRNA sequence of the GNPTAB gene is taken
- Page 8, line 256 - please correct the unit notation
- Page 8, line 263 - please elaborate on the abbreviation NICU
- Page 8, line 268 - please elaborate on the abbreviation CT in brackets
- Page 8, line 291 and 293– it is: ‘showed elevated dermatan sulfate, ….’ please specify elevated level, concentration…
- Page 9-10 – table 4,5 and 6 - please rearrange the table so that the same values for physiological norms do not appear several times in the table - perhaps it would be worthwhile to list them in a footnote under the table - this would make the table itself more readable
- Page 9-10, table 5 and 6 - please share the same unit of concentration for all enzymes - there is no need to multiply the same information in each column
- Page 10, line 317 - double ‘in’ in the notation
- Page 10-11, table 7 and 8 - please consider changing the first line of the table with ‘neaonatal samples: MLII vs HN' to ‘upregulated proteins in ML II patients, neonatal samples’ (table 7) and ‘upregulated proteins in healthy neonates, neonatal samlpes’ (table 8) -or something like that - in the proposed version, at first glance the tables suggest the same content -
- Page 13-14, table 9 and 10 - as above, please change first line to be more precise
- Page 15-16. Table 11 and 12 - as above
- Page 16, table 13 - table 13 needs to be reorganised:
1) the description of the table is not precise, neither is the table itself - according to the description in the text of the publication, small proteins dysregulated in both patients and healthy neonates are to be found here
2) the first two lines (proteins designated as Q01955 and O43451 - found earlier in Tables 7, 9 and 12 upregulated proteins in MLII specimens (neonates and diagnostic samples) - so in total this is a repetition from Table 12
3) three further proteins which are found in earlier Tables 8, 10 and 11 and are therefore upregulated in ML II neonates and downregulated in ML II diagnostic samples, the summary Table 11 indicates that considering both groups of samples (neonates and diagnostics) these proteins are however downregulated - please explain + again repeat from Table 11
4) the last group of proteins preceded by a line with the description ‘neonatal samples vs diagnostic samples remain elevated’ - contains only proteins previously listed in Table 7 and therefore upregulated proteins found only in the ML II neonates population of samples - please clarify, correct or delete the table as it duplicates information already appearing earlier
5) in addition, in the table there is a column described as protein code and instead of protein code uniprot code is listed
- Page 19, line 479 - please delete repetition
- Page 19, line 481 – please write tandem mass spectrometry (MS/MS) techniques instead MS/MS techniques
- Page 21, line 557 – Some data is missing in the reference - please complete
It is worth noting further that of the 42 articles cited in the references section, 19 items are from the last five years and no excessive self-citation was found.
Author Response
REVIEWER 1
Comments and Suggestions for Authors
In this paper, the authors characterize the clinical, biochemical and proteomic profiles of three patients with a confirmed diagnosis of mucolipidosis type II, followed up for 24 months. This work is interesting and informative. The text of the article is written in a way that enables the authors' argument to be followed comprehensibly. However, the introduction, discussion and conclusion section of the paper should be a bit more refined to make it a smoother read, e.g. in my subjective opinion the last paragraph of introduction should be one position earlier etc. In addition, there are paragraphs in the discussion which are more suited to the introduction (page 18, lines 427-430), or paragraphs which duplicate information already present (e.g. page 18, lines 242-226) and do not constitute a proper discussion of the results obtained. Figure 7 should also be placed earlier in the text and so on.
ANSWER. Thank you for your feedback. We have moved Figure 7 closer in the text so that it is now called Figure 3 and added the following text.
Figure 3. Biochemical routes for degradation of Heparan sulfate and Dermatan sulfate. Enzymes analyzed: IDS - iduronidate 2-sulfatase, IDUA - alpha iduronidase; ARSB - arylsulfatase B, NAGLU alpha-N-acetyl glucosaminidase; GUSB - beta glucuronidase. Enzymes not analyzed: Heparanase- HSPE, N-sulfoglucosamine sulfohydrolase- SGSH, N-acetylglucosamine-6-sulfatase- GNS, Hyaluronidase- HYAL.
- Page 2, line 46 – it is: ‘MIM#252500 and MIM#252600 - shouldn't ‘MIM’ be ‘OMIM’ instead?
ANSWER. That information has been deleted of the text “and III α/β (MIM#252600)”
- Page 2, line 48 - the Enzyme Commission (EC) number of N-acetylglucosamine-1-phosphotransferase should be added. The same recommendation is valid for the other enzymes whose names appear for the first time in the whole text (e.g. lines 109-113)
ANSWER. Add Enzyme Commission (EC) number 2.7.8.17 for GlcNAc-PTase acid sphingomyelinase (ASMD) EC 3.1.4.12, alpha iduronidase (IDUA) EC 3.2.1.76, iduronidate 2-sulfatase (IDS) EC 3.1.6.13, alpha-N-acetyl glucosaminidase (NAGLU) EC 3.2.1.50, beta-glucuronidase (GUSB) EC 3.2.1.31, arylsulfatase B (ARSB) EC 3.1.6.12.
Alpha- galactosidase (GAA) (3.2.1.22)
galactocerbrosidase -GALC (EC 3.2.1.46)
alpha glucosidase -GLA (EC 3.2.1.20)
beta glucosidase -GBA (3.2.1.21)
galactose 6-sulfatase -GANLS (EC 2.5.1.5)
beta galactosidase -GLB1 (EC 3.2.1.23)
EC include in abbreviation list
- Page 2, line 48 – please elaborate on the abbreviation M6P
ANSWER. M6P mannose 6-phosphate included in abbreviation list.
- Page 2, line 59 - please add the date of access
ANSWER. Date of access added.
www.orpha.net, accessed 3 Jan. 2025)
- Page 3, line 94 and 95 - two different ranges of norms are assigned to the same parameter - IgM - please correct
ANSWER. IgM reference values vary depending on the age of the patient. The reference values are shown in section 5 (IgM 3-6 months 5-50 mg/dl, 6-12 months 8-70 mg/dl, 12-24 months 10-100 mg/dl).
- Page 3, line 100 - the elaboration of the ‘WES’ abbreviation can be found in the following text (line 202) and the abbreviation itself appears here for the first time - please correct
ANSWER. WES whole-exome sequencing included in abbreviation list.
- Page 3, line 109 – it is ‘using MS/MS – it should rather be: ‘using liquid chromatography-tandem mass spectrometry (LC-MS/MS)’
ANSWER. We changed the acronym MS/MS to LC-MS/MS for the determination of GAG and enzymatic assays and Proteomic.
Page 3, line 118 - acronyms such as HS, DS should appear early, in line 116
ANSWER. Abbreviations have been checked and modified thought all the text and table 2.
- Page 3, line 123 - add city and state information for Revvity to be consistent with earlier text (e.g. line 101)
ANSWER. Add in the text: Revvity (Waltham, Massachusetts, U.S.) Page18 line 472
- Page 4, line 155 – it is: ‘MeOH/ChCl3’ - it should be: ‘MeOH/ CHCl3’
ANSWER. Add in the text: MeOH/CHCl3 Page 18, line 504
- Page 4, line 156 – RC-DC kit - full kit name not added - please complete
ANSWER. Add in the text: Protein Assay (reducing agent and detergent compatible) (BioRad). Page 18, line 506
- Page 4, line 164 - please add a specific literature reference
ANSWER. The references are 28,38–40
- Page 6, line 204 – add the name of the database from which the ID for the mRNA sequence of the GNPTAB gene is taken
ANSWER. Text added: GNPTAB (NM_024312.5). This information has been communicated to the Clin Var database (SUB15044432). Page 3 Line 109
- Page 8, line 256 - please correct the unit notation
ANSWER. Add in the text: gestational age (GA). Page 3 line 102
- Page 8, line 263 - please elaborate on the abbreviation NICU
ANSWER. NICU (Neonatal intensive care unit). Page 5 line 173
- Page 8, line 268 - please elaborate on the abbreviation CT in bracke
ANSWER. CT (Computed Tomography) scan. Page 5 line 178
- Page 8, line 291 and 293– it is: ‘showed elevated dermatan sulfate, ….’ please specify elevated level, concentration
ANSWER. The information is shown in table nº 2, along with reference values and concentration.
- Page 9-10 – table 4,5 and 6 - please rearrange the table so that the same values for physiological norms do not appear several times in the table - perhaps it would be worthwhile to list them in a footnote under the table - this would make the table itself more readable
ANSWER. Thank you very much for your comment. Reference values for enzymatic activities and GAG quantification appear several times in the table to make it easier for the reader to identify the altered values. Reference values ​​are not the same for the samples of Neonates as for the other ages.
- Page 9-10, table 5 and 6 - please share the same unit of concentration for all enzymes - there is no need to multiply the same information in each column
ANSWER. Tables have been modified to reduce the same redundant information (concentration values in each column).
- Page 10, line 317 - double ‘in’ in the notation
ANSWER. We deleted it
- Page 10-11, table 7 and 8 - please consider changing the first line of the table with ‘neaonatal samples: MLII vs HN' to ‘upregulated proteins in ML II patients, neonatal samples’ (table 7) and ‘upregulated proteins in healthy neonates, neonatal samlpes’ (table 8) -or something like that - in the proposed version, at first glance the tables suggest the same content
ANSWER. Change in the text: TABLE 5: Upregulated proteins in ML II neonatal patients Vs Healthy neonatal samples
Change in the text: TABLE 6 : Downregulated ML II neonatal patients Vs Healthy neonatal samples
- Page 13-14, table 9 and 10 - as above, please change first line to be more precise
ANSWER. Change in the text: TABLE 7: Upregulated proteins in ML II diagnosis patients Vs Healthy neonatal samples
Change in the text: TABLE 8: Downregulated ML II neonatal patients Vs Healthy neonatal samples
- Page 15-16. Table 11 and 12 - as above
ANSWER. Change in the text: TABLE 9: Upregulated proteins in healthy neonates vs samples of ML II patients (neonatal period and diagnosis)
Change in the text: TABLE 10: Downregulated protein found in ML II patients (neonatal period and at diagnosis) vs healthy neonates
- Precise Page 16, table 13 - table 13 needs to be reorganized
ANSWER. Thank the review for this comment;
Change in the text: TABLE 11:
Upregulated proteins in Neonatal and diagnosis ML samples VS healthy neonates
Downregulated proteins in healthy neonates VS Neonatal and diagnosis ML samples
Upregulated proteins in Neonatal ML samples vs Diagnostic ML samples remain elevated
- The description of the table is not precise, neither is the table itself - according to the description in the text of the publication, small proteins dysregulated in both patients and healthy neonates are to be found here.
ANSWER. Thank you for your comment. You are right. We had a mistake in table 11 because the statement was reversed
- the first two lines (proteins designated as Q01955 and O43451 - found earlier in Tables 7, 9 and 12 upregulated proteins in MLII specimens (neonates and diagnostic samples) - so in total this is a repetition from Table 12. the first two lines (proteins designated as Q01955 and O43451 - found earlier in Tables 7, 9 and 12 upregulated proteins in MLII specimens (neonates and diagnostic samples) - so in total this is a repetition from Table 12.
ANSWER. Thank you for your comment. You are right. We had a mistake in table 11 because the statement was reversed
- three further proteins which are found in earlier Tables 8, 10 and 11 and are therefore upregulated in ML II neonates and downregulated in ML II diagnostic samples, the summary Table 11 indicates that considering both groups of samples (neonates and diagnostics) these proteins are however downregulated - please explain + again repeat from Table 11
ANSWER. Thank you for your comment. You are right. We had a mistake in table 11 because the statement was reversed
- the last group of proteins preceded by a line with the description ‘neonatal samples vs diagnostic samples remain elevated’ - contains only proteins previously listed in Table 7 and therefore upregulated proteins found only in the ML II neonates population of samples - please clarify, correct or delete the table as it duplicates information already appearing earlier
ANSWER. Thank the review for the comment; we change the table’s text in order to make these tables easier to understand. The last tables are a summary of the most important proteins we found deregulated
- in addition, in the table there is a column described as protein code and instead of protein code uniprot code is listed
ANSWER. Add the table protein code
Upregulated proteins in Neonatal samples and diagnosis samples VS Wt |
||||
Code Protein |
Protein Code |
Protein |
FC Neo |
FC Dg |
Q01955 |
CO4A3 |
Collagen alpha-3(IV) chain |
2,13 |
2,34 |
O43451 |
MGA |
Maltase-glucoamylase, intestinal |
1,65 |
1,61 |
Downregulated proteins in Wt VS Neonatal samples and diagnosis samples |
||||
Code Protein |
Protein Code |
Protein |
FC Neo |
FC Dg |
Q15181 |
IPYR |
Inorganic pyrophosphatase |
0,58 |
0,71 |
P26639 |
SYTC |
Threonine--tRNA ligase, cytoplasmic |
0,27 |
0,50 |
P07686 |
HEXB |
Beta-hexosaminidase subunit beta |
0,31 |
0,31 |
Upregulated proteins in Neonatal samples vs Diagnostic samples remain elevated |
||||
Code Protein |
Protein Code |
Protein |
FC Neo |
FC Dg |
P30711 |
GSTT1 |
Glutathione S-transferase theta-1 |
4,38 |
|
P63241 |
IF5A1 |
Eukaryotic translation initiation factor 5A-1 |
3,22 |
|
Q99715 |
COCA1 |
Collagen lpha-1(XII) chain |
3,24 |
|
Q16775 |
GLO2 |
Hydroxyacylglutathione hydrolase, mitochondrial |
1,90 |
|
Q9NZI8 |
IF2B1 |
Insulin-like growth factor 2 mRNA-binding protein 1 |
2,18 |
|
P08779 |
K1C16 |
Keratin, type I cytoskeletal 16 |
2,05 |
|
- Page 19, line 479 - please delete repetition
ANSWER. We delete the repetition
- Page 19, line 481 – please write tandem mass spectrometry (MS/MS) techniques instead MS/MS techniques
ANSWER. Change in the text all MS/MS by LC-MS/MS
- Page 21, line 557 – Some data is missing in the reference - please complete
ANSWER. Thank you for your comment. We complete the reference.
Mol Cell Proteomics. 2011 Jan;10(1):M110.003335. doi: 10.1074/mcp.M110.003335
It is worth noting further that of the 42 articles cited in the references section, 19 items are from the last five years and no excessive self-citation was found.
Thank you very much for your comment.
Unfortunately, there is not much bibliography on this disease because not much is known about it. We are working to increase this information.

Reviewer 2 Report
Comments and Suggestions for Authors
I had the pleasure of reviewing the work of Monteagudo-Vilavedra and colleagues regarding the phenotypic and biochemical profiling of patients with mucolipidosis type II. Overall, this is a very well-written and comprehensive manuscript.
Major comment: I believe there is an excess of figures and tables, and I am not convinced of the importance of the STRING interaction studies. I would suggest mentioning the interactions in the text to support the SWATH analysis, for example, instead of presenting them in detail.
Minor comments:
Figure 3 is hard to read due to the size of the images. Additionally, it might be better to mention the STRING images in the text only.
Figure 7 could be moved to the Results section, and perhaps it would be helpful to indicate which patient had each finding.
I hope my suggestions help improve the readability of this excellent manuscript.
Author Response
REVIEWER 2
Comments and Suggestions for Authors
I had the pleasure of reviewing the work of Monteagudo-Vilavedra and colleagues regarding the phenotypic and biochemical profiling of patients with mucolipidosis type II. Overall, this is a very well-written and comprehensive manuscript.
Major comment: I believe there is an excess of figures and tables, and I am not convinced of the importance of the STRING interaction studies. I would suggest mentioning the interactions in the text to support the SWATH analysis, for example, instead of presenting them in detail.
ANSWER. We have removed the STRIG images and left the volcano plot as figure 4.
We deleted this text:
Using string tool to analyze the upregulated proteins in the ML II patient’s vs neonatal healthy we can see proteins related to collagen containing extracellular matrix (red balls), extracellular exosome (yellow), extracellular space (green), extracellular region (blue) (figure 3B and C). Among these proteins we interestingly found AACT, MMP8 and CATG which are related with matrix degradation or ODPB that is the most upregulated and is related to Krebs cycle.
Minor comments:
Figure 3 is hard to read due to the size of the images. Additionally, it might be better to mention the STRING images in the text only.
Figure 7 could be moved to the Results section, and perhaps it would be helpful to indicate which patient had each finding.
I hope my suggestions help improve the readability of this excellent manuscript.
ANSWER. Thank you for your feedback. We have moved Figure 7 closer in the text so that it is now called Figure 3 and added the following text.

Round 2
Reviewer 1 Report
Comments and Suggestions for Authors
The authors responded accordingly. I recommend the work for publication with two minor corrections: 1) Page 46, line 45-46: It is” (…)GNPTAB gene producing N-acetylglucosamine-1-phosphotransferase (GlcNAc-PTase) deficiency. Enzyme Commission (EC) number 2.7.8.17), I recommend: ‘(…)GNPTAB gene producing N-acetylglucosamine-1-phosphotransferase (GlcNAc-PTase, EC 2.7.8.17) deficiency’ and 2) Table 2,3,4 - please list the physiological norms of the relevant parameters under the tables containing them, otherwise the same text is repeated several times in the tables.
Author Response
two minor corrections:
- Page 46, line 45-46: It is” (…)GNPTAB gene producing N-acetylglucosamine-1-phosphotransferase (GlcNAc-PTase) deficiency. Enzyme Commission (EC) number 2.7.8.17), I recommend: ‘(…)GNPTAB gene producing N-acetylglucosamine-1-phosphotransferase (GlcNAc-PTase, EC 2.7.8.17) deficiency’
Answer. Thanks for your comment I correct the sentence.
(GlcNAc-PTase, EC 2.7.8.17)
- and 2) Table 2,3,4 - please list the physiological norms of the relevant parameters under the tables containing them, otherwise the same text is repeated several times in the tables.
Answer. Thanks for your comment I added in the text.
High values ​​are marked in bold
Thank you so much for you suggestions